

# Effects of different grazing intensities on plant species diversity at different spatial scales in a desert steppe in Inner Mongolia

Changlin Xue[1], Shijie Lv[2], Yanling Wu[1], Jie Yun[1], Rui Dong[1] and Wentao Wang[1]

[1] College of Life Science and Technology, Inner Mongolia Normal University, Hohhot, Inner Mongolia Autonomous Region, China
[2] College of Grassland, Resources and Environment, Inner Mongolia Agricultural University, Hohhot, Inner Mongolia Autonomous Region, China

## ABSTRACT

The effect of grazing intensity on plant diversity has been widely studied. In this study, desert steppes with different grazing intensities (no grazing (CK), light grazing (LG), moderate grazing (MG), heavy grazing (HG), and extremely heavy grazing (EG)) in Inner Mongolia were selected to study the changes in species diversity at different spatial scales ($\alpha$, $\beta$, and $\gamma$ diversity) and the $\alpha$ diversity of different plant groups (dominant species, common species, and rare species).The results showed that the $\alpha$, $\beta$, and $\gamma$ diversity first decreased and then increased with increasing grazing intensity, and $\beta$ diversity was observed to be the most sensitive index to the grazing intensity. Grazing had the greatest impact on the $\alpha$ diversity of rare species and the least impact on the $\alpha$ diversity of common species. Therefore, rare species are of great significance for the maintenance and assessment of biodiversity. We concluded that grazing leads to a sensitive response of $\beta$ diversity, and this sensitive phenomenon is mainly affected by rare species. The results could provide scientific bases for the protection of plant diversity and sustainable grazing in desert steppes.

## INTRODUCTION

Grassland constitutes an important part of terrestrial ecosystems (*Bengtsson et al., 2019*). The desert steppe is a large component of the semi-arid grassland ecosystem in northern China, and provides significant resources for livestock production, and is particularly vulnerable to desertification. Desertification is one of the most serious types of land degradation, and overgrazing is regarded as one of the main causes for the degradation and desertification of desert steppe over the recent decades (*Ha-Lin et al., 2009*; *Mei et al., 2021*; *Ruiyang et al., 2018*). Overgrazing alters plant community composition, soil erosion and biodiversity (*Cathleen Maria et al., 2016*; *Havstad, Herrick & Schlesinger, 2000*; *Wen et al., 2023*). At present, considerable attention is paid to how to reasonably use grassland resources by grazing, *e.g.*, by evaluating the effects of different grazing intensities

Corresponding author
Yanling Wu, wuyanling@imnu.edu.cn

and methods on grassland productivity, plant species composition and diversity, and soil properties (*Deng, Sweeney & Shangguan, 2013*; *Wen et al., 2023*). These studies have enhanced our comprehension of the potential influences of grazing. As the intermediate disturbance hypothesis states, biodiversity is highest under moderate grazing intensity disturbance (*Michael, 2014*). The "dynamic equilibrium hypothesis" is different from this view. While focusing on grazing intensity, it also takes into account the control of competition, predation and productivity on biodiversity (*Huston, 1979*). Jeremy W. Fox also proposed that a study of more than 100 published articles on the diversity-disturbance relationship found that the probability of the diversity peak occurring under moderate disturbance was less than 20% (*Jeremy, 2012*). Therefore, understanding and revealing the impacts that grazing exerts on grassland biodiversity and ecosystem functions are of great significance for desertification control, and the sustainable development of grassland biodiversity.

Overgrazing is one of the key drivers of decrease in grassland biodiversity (*Péter et al., 2018*). Grazing can affect ecosystem processes and biodiversity, altering the spatial heterogeneity of vegetation (*Adler, Raff & Lauenroth, 2001*). The variations in three components, including intra-community or α diversity, inter-community or β diversity, and intra-regional or γ diversity, affect the global taxonomic richness (*Sepkoski, 2016*). The α, β, and γ components of biodiversity can be used to evaluate and monitor the impacts of human activities on biodiversity (*Halffter, 1998*). The α diversity refers to diversity on a local scale. For plants, α diversity is often equated to the count of species identified during the inventory of a vegetation plot of defined size (*Tobias et al., 2022*; *Rasmus et al., 2016*). β diversity refers to the dissimilarity in species composition among sites along environmental gradients (*Yang et al., 2014*). Understanding spatial variation in biodiversity along environmental gradients is a central theme in ecology, and differences in β diversity are often used to infer variation in the processes structuring communities (*Kraft et al., 2011*). Exploring intra-community (α) and inter-community (β) diversity changes has special relevance, since only their joint evaluation can reveal unequivocally how anthropogenic stressors influence spatial patterns in biodiversity (*Jani, 2011*). *Whittaker (1972)* and *Whittaker (1977)* proposed that α and β diversity have a combined impact on γ diversity that describes the species richness of a landscape (*Whittaker, 1972*; *Whittaker, 1977*). This indicates the higher versatility of the spatial γ scale, that can be decomposed into the α component within ecosystems and the β component across ecosystems, and correlation functions can also be corrected (*Anne et al., 2023*; *Halffter, 1998*; *Marília Maria Silva da & Fernando Augusto, 2022*).

The desert steppe is the transition region between the desert and traditional grassland steppes on the Inner Mongolia plateau (*Mei et al., 2021*). Desertification is one of the most serious environmental and socio-economic problems in many arid and semi-arid regions of the world (*Gomes et al., 2003*; *Javier, Jaime & Susanne, 2007*), and grazing is considered the main anthropogenic factor driving the plant community structure and ecological functions of the desert steppe (*Jiankang et al., 2019*). In previous studies, in which the effects of grazing intensity on species diversity in the desert steppe were investigated, the focus was mainly on the α diversity, but limited studies have been carried out on the β and
γ diversity, as well as the diversity of plant groups, including dominant species, common species, and rare species. To address these research gaps, the present study was conducted in the desert steppe of Harden Hushu Gacha, Saihantala Town, Sonid Right Banner, Xilin Gol League, Inner Mongolia Autonomous Region (IMAR), China, with the aim of answering the following questions: (1) How differently does the α diversity of different plant groups respond to various grazing intensities? (2) How does the plant species diversity at different spatial scales (α, β, and γ diversity) respond to the increase in grazing intensity? And (3) What are the corresponding relationships between different grazing intensities and plant species diversity? The solution to these relevant problems can provide theoretical and practical support for the sustainable development of desert steppes and the construction of grassland ecology.

## MATERIALS & METHODS

### Sample site overview and experimental design

The experimental site is located in Haden Hushu Gacha Town, Saihantala Town, Sunit Right Banner, Xilin Gol League, Inner Mongolia Autonomous Region (IMAR), China, at 41°55′∼43°39′N latitude, 111°08′∼114°16′E longitude, with an average altitude of 1,150.8 m, and a linear distance of 50.81 km from the banner set up by the government of China. This area has a typical arid temperate continental climate, with cold winters and hot summers. The highest yearly average temperature is 38.7 °C, while the lowest is −38.8 °C, with a large difference in the annual average temperature. The frost-free period is 130 days. From 2000 to 2020, the recorded average annual precipitation was 199.6 mm, while the average annual evaporation was 2,384 mm, and the average speed of prevailing westerly wind was 5.5 m/s. The study site is a semi-desert grassland region situated in the transition zone between steppe and desert, and has mainly brown soil rich in calcium.

The study area is dominated by herbaceous plants such as *Stipa breviflora*, *Cleistogenes songorica*, and *Allium polyrhizum*.

The fence grazing was adopted in this research, and three experimental blocks were randomly set up, with each block having five plots of different grazing intensities. There were five different treatments of grazing intensity (Fig. 1), including no grazing (control) (CK), light grazing (LG), moderate grazing (MG), heavy grazing (HG), and extremely heavy grazing (EG), with 0, 1.54, 1.92, 2.31, and 2.69 sheep units. hm$^{-2}$, respectively, grazing intensities were established with 0, 4, 5, 6 and 7 sheep per treatment. Each grazing plot had an area of about 2.6 hm$^2$ from the total area of approximately 39.2637 hm$^2$. The grazing treatments were applied from early May to late October (*Xing et al., 2021*). No permits were needed at any locations.

### Sampling design

The sampling points were set up using random distribution. There was a total of 15 plots, with each plot being 1 m ×1 m with three samples. The number of occurrences of plants was recorded based on species. Based on the vegetation types (*Stipa breviflora + Cleistogenes songorica +Allium polyrhizum*) in the study area and the importance value (IV) data obtained from 45 sampled plots, plant species were categorized into three plant groups:
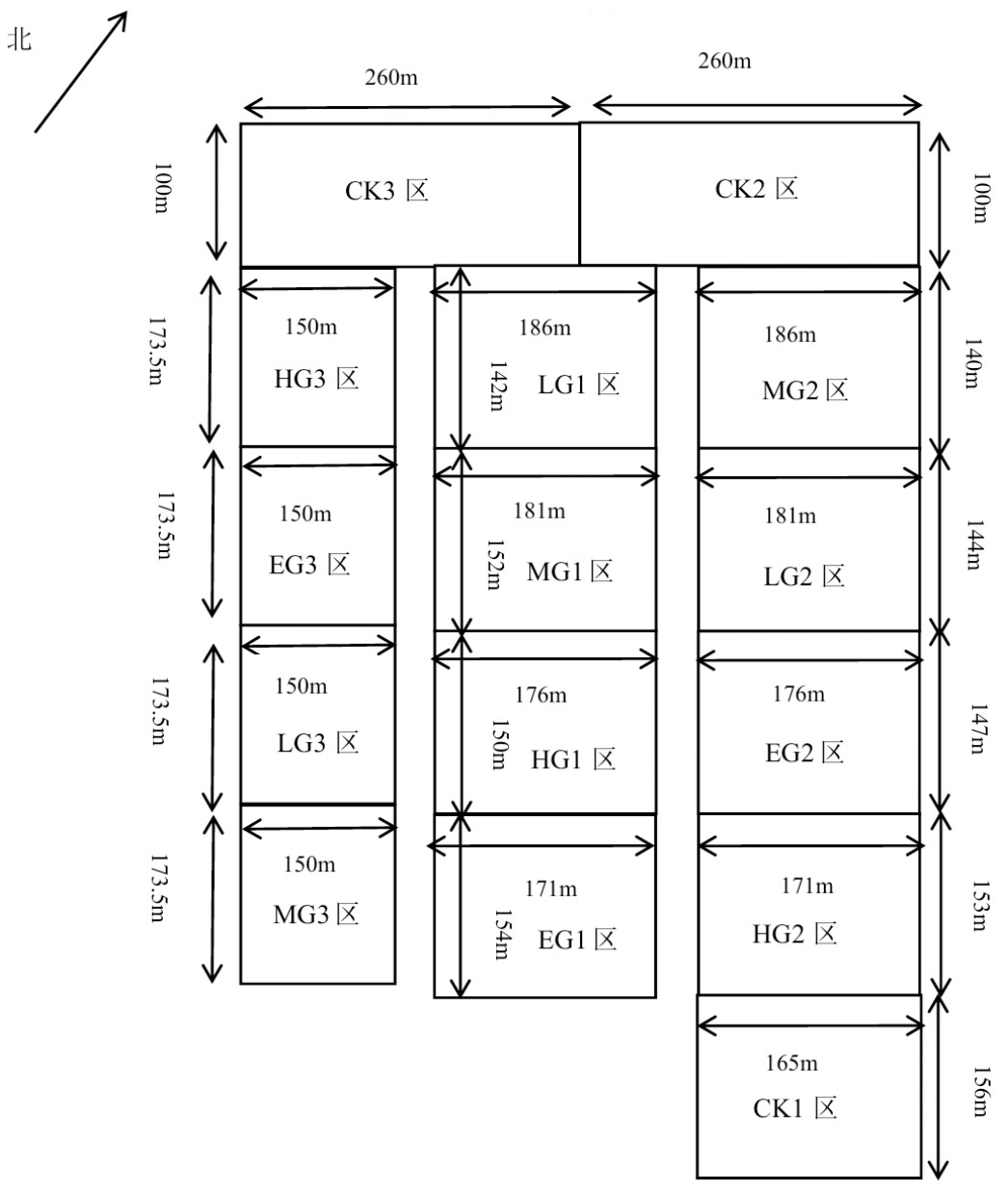

**Figure 1  Plot layout.**

dominant species, common species, and rare species (*Pan, Liu & Niu, 2018*; *Yang et al., 2014*). *Stipa breviflora*, *Cleistogenes songorica*, and *Allium polyrhizum* were the dominant species in the study area. There were, however, seven common species such as *Convolvulus ammannii*, *Neopallasia pectinata*, *Setaria viridis, etc.*, and *Asparagus cochinchinensis*, *Allium mongolicum*, and *Allium tenuissimum* were among the seven rare species (Table 1).

**Table 1** Classification of the dominant, common and rare species.

| Classification of species | The Latin name of species | Abbreviations of species names in Latin |
|---|---|---|
| Dominant species | *Stipa breviflora* Griseb. | *S.bre* |
| | *Cleistogenes songorica* (Roshev.) Ohwi | *C.son* |
| | *Allium polyrhizum* Turcz. ex Regel | *A.pol* |
| Common species | *Convolvulus ammannii* Desr. | *C.amm* |
| | *Neopallasia pectinata* (Pall.) Poljak. | *N.pec* |
| | *Setaria viridis* (L.) Beauv. | *S.vir* |
| | *Eragrostis pilosa* (L.) Beauv. | *E.pil* |
| | *Kochia prostrata* (L.) Schrad. | *K.pro* |
| | *Carex duriuscula* subsp. *stenophylloides* | *C.dur* |
| | *Artemisia capillaris* Thunb. | *A.cap* |
| Rare species | *Asparagus cochinchinensis* (Lour.) Merr | *A.coc* |
| | *Astragalus galactites* Pall. | *A.gal* |
| | *Oxytropis racemosa* Turcz. | *O.rac* |
| | *Allium tenuissimum* L. | *A.ten* |
| | *Allium mongolicum* Regel | *A.mon* |
| | *Cleistogenes squarrosa* (Trin.) Keng | *C.squ* |
| | *Caragana stenophylla* Pojark. | *C.ste* |

## Data processing and analysis

The plant species diversity of a desert steppe under different grazing intensities could be calculated using the following formulas: (1) $\alpha$ diversity index: $\alpha = \frac{1}{3}\sum_{i=1}^{3} k_i$, where: $k_i$ represents the sum of species in the $i$th 1 m $\times$ 1 m quadrat; (2) $\gamma$ diversity index: $\gamma =$ the total number of species recorded under each grazing intensity; (3) $\beta$ diversity index: $\beta = \gamma$-$\alpha$; (4) Importance Value (*IV*): $IV =$ (relative height + relative cover + relative density)/3 $\times$ 100%.

## Statistical analysis

All statistical analyses were performed using R-Studio software. The ggplot2 package was used to create the percent stacked bar charts to analyze the responses of different plant community-type frequencies to grazing intensities. The differences in the $\alpha$ diversity of dominant species, common species, and rare species among different grazing intensities were determined by the one-way analysis of variance (ANOVA), and bar graphs were plotted using the ggplot2 and ggsignif packages. Thereafter, the regression analysis was used to evaluate the relationships between five grazing intensities (CK, LG, MG, HG, and EG) taken as continuous variables and diversity indices by linear, quadratic, and exponential functions. The optimal model-fitting was implemented using $R^2$ and *P* values, and the ggplot2 and ggpmisc packages were used to draw the fitted curves. Finally, after averaging the obtained values for diversity indices, we employed the FactoMineR and CA packages in R to perform the corresponding analysis to explore the relationships among different grazing intensities and species diversity indices ($\alpha$, $\beta$, $\gamma$, DA, CA, and RA).

## RESULTS

### Response of species frequency to grazing intensity

The dominant species, including *S. breviflora* and *C. songorica*, showed high grazing tolerance, which first increased and then stabilized with the increase in grazing intensity. In contrast, the grazing tolerance of *A. polyrhizum* is relatively weak, and it belongs to the plant species that are sensitive to grazing. Its frequency decreases as the grazing intensity increases (Fig. 2A). Common species generally exhibited neutral responses to grazing, *e.g.*, changes in grazing intensity hardly resulted in the changing trends of the species frequency; the trends were, however, almost constant or fluctuating (Fig. 2B). Rare species were generally sensitive to grazing. That is, grazing can cause a sharp decline in their frequency (such as *Allium mongolicum*) or a trend of first increasing and then decreasing (such as *Caragana stenophylla*), but for Asparagus cochinchinensis, the changing patterns were not obvious, and thus, it can be classified as a grazing-insensitive species (Fig. 2C).

### Response of alpha diversity of dominant species, common species and rare species to grazing intensity

The strongest response to different grazing intensities was achieved by the α diversity of rare species, followed by that of dominant species; the α diversity index of common species, however, was the least sensitive to grazing intensity (Fig. 3). The dominant species displayed the α diversity index ranging from 1.33 to 2.67, with a significant difference detected between the heavily grazed areas and the ungrazed (control) areas ($P < 0.05$). In addition, the α diversity of common species was between 3.67 and 5.33, but no significant difference was noted under different grazing intensities ($P > 0.05$). Rare species showed α diversity within the range of 0.33 to 2.33, and a changing trend of first decreasing and then stabilizing with the increase in grazing intensity, and different grazing intensities differed significantly ($P < 0.05$).

### Response of α, β and γ diversity to grazing

With increased grazing intensity, all three α diversity, β diversity, and γ diversity underwent an initial decrease, followed by an increase. As shown in Fig. 4, following a downward trend of diversity indices in the CK and different grazing treatments, α and γ diversity indices started increasing slightly in the extremely heavy grazing treatment, while the β diversity index began increasing from the heavy grazing (HG) to EG. This indicates that the optimal grazing intensity was below the MG intensity. Compared with the control treatment, α, β, and γ diversity decreased by 13%, 66%, and 26% under light grazing, whereas under moderate grazing, decreases by 22%, 69%, and 34% were recorded, respectively. Moreover, when grazing was heavy, decreases of 28%, 62%, and 37% were obtained, and extremely heavy grazing caused decreases by 22%, 48%, and 29% in α, β, and γ diversity, respectively. Thus, β diversity was more sensitive to grazing, whereas α diversity was least responsive to it.

### Correspondence between grazing intensity and species diversity

The first two principal components (PCs) could explain more than 97% of the variance in original data (Fig. 5), and the goodness-of-fit was very high, indicating that the correlations

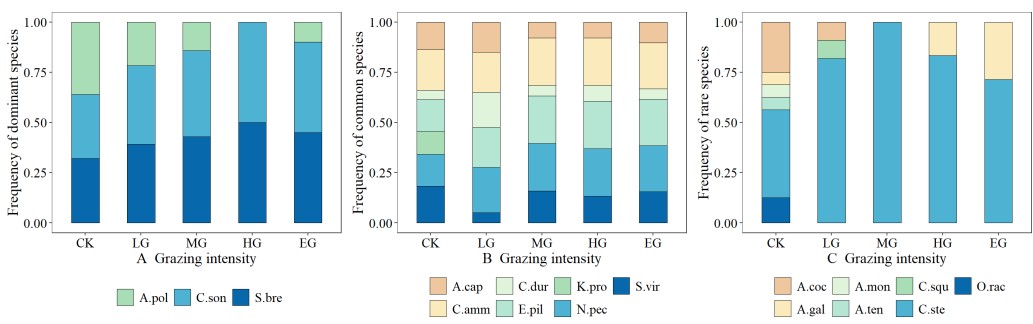

**Figure 2** Effects of grazing intensity on the frequency of dominant species, common species and rare species.

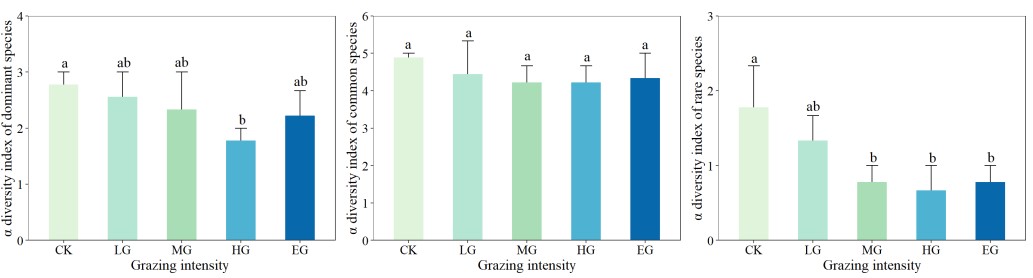

**Figure 3** Effects of grazing intensity on α diversity of dominant species, common species and rare species.

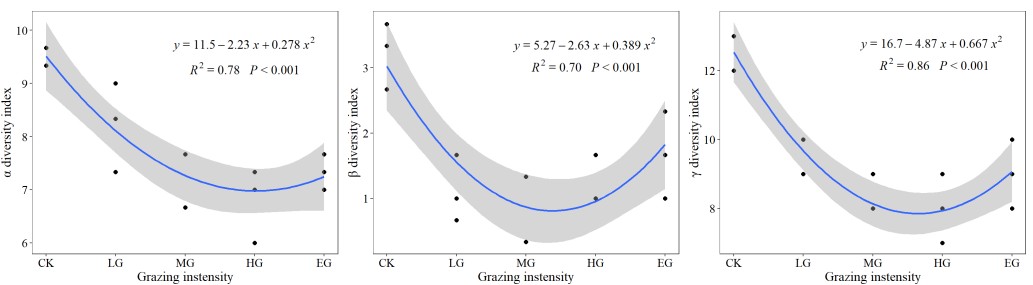

**Figure 4** The responses of α, β, and γ diversity to grazing intensity.

existed among grazing intensities and diversity indices. At the same time, CK, LG, and MG were found to be above the principal component horizontal axis (Dim1), while HG and EG were below it, which demonstrates that the effects of HG and EG treatments on the species diversity of the plant community were different from those of CK, LG, and MG, which were arranged from left to right along the horizontal axis. These results revealed that there was a linear variation in the species diversity of the plant community under grazing intensities below MG. The α diversity of dominant species (DA) and common species (CA), and the α and γ diversity of the plant community were insensitive to grazing

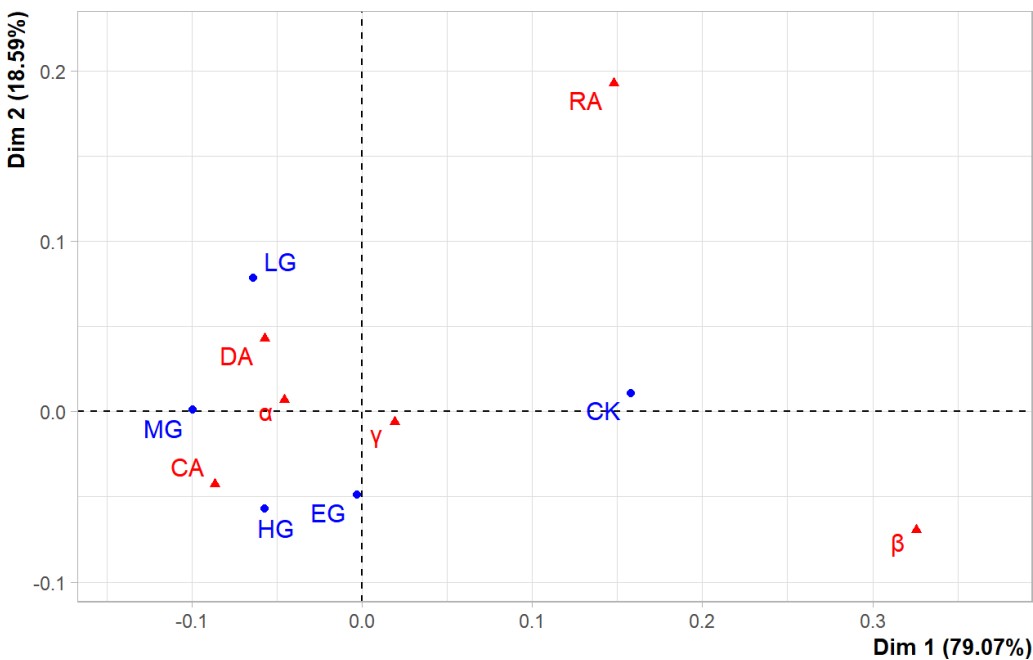

**Figure 5** The correspondence analysis results.

intensity, and therefore, they were positioned in close proximity to LG, MG, HG, and EG. However, the α diversity of rare species (RA) and the β diversity of the plant community were sensitive to grazing, but they showed nonlinear relationships with different grazing intensities, and thus, they were far away from each grazing intensity point along the axis.

## DISCUSSION

### Effects of grazing on plant community species diversity

Grazing represents the most extensive land utilization mode worldwide. Grazing is one of the main factors affecting plant diversity (*Fernando et al., 2022*; *Jinlan et al., 2022*). In the study area located in a desert steppe, the α and γ diversity indices under the light, moderate, and heavy grazing decreased compared with those in the non-grazed areas (control), indicating that grazing would reduce plant diversity. Heavy grazing should be avoided to protect the diversity and stability of grassland ecosystems, which is different from the intermediate disturbance hypothesis (*Michael, 2014*). The research results obtained by *Takehiro et al. (2007)* also showed that the plant species diversity in a desert steppe could not support the intermediate disturbance hypothesis (*Takehiro et al., 2007*), but were consistent with the "dynamic equilibrium model", which predicts that competition, predation, and productivity together contribute to controlling species diversity (*Huston, 1979*; *Michael, 2014*). The α and γ diversity indices increased under extremely heavy grazing compared to those in the heavily grazed site, which is due to the degradation of vegetation communities by grazing, especially for more biennial plants but fewer perennials (*Yousif*

*Mohamed et al., 2020*; *Zhen'an et al., 2016*), and despite the increase in species diversity, these communities remained in a degraded state.

In the ungrazed, lightly grazed, and moderately grazed areas, the linear decrease in the β diversity index with the increase in grazing intensity was observed. Compared with the control treatment, the β, α, and γ diversity indices decreased by 69%, 21%, and 34%, respectively, indicating the highest sensitivity of the β diversity to the grazing intensity, followed by γ diversity and α diversity. The increase in grazing intensity caused decreases in herbivore selectivity (*Edina et al., 2016*; *Péter et al., 2018*), and the gradual disappearance of grazing-sensitive species, such as *Kochia prostrata* and *Allium tenuissimum*, occurs (*Milchunas, Sala & Lauenroth, 1988*), resulting in the decrease in the γ diversity index (*Julia et al., 2021*). The α diversity index, which is a measure of the average number of species within the community (*Maria Teresa, Francesc & Enric, 2021*), was less affected by grazing intensity. Therefore, the α and γ diversity together induced the sensitivity of the β diversity to grazing intensity.

**Effects of plant taxa on species diversity under grazing conditions**

At the level of community type, the α diversity of rare species was the most responsive index to changes in grazing intensity (*Jiaojiao et al., 2024*), followed by that of dominant species, while common species were the least sensitive in terms of alpha diversity. The possible reason is that most of the rare species are grazing-sensitive and highly palatable species (*Shijie et al., 2024*), such as *Allium tenuissimum*, *Allium mongolicum*, *etc.*, that began to decrease or even disappear in light grazing areas (*Shuxia et al., 2010*). Furthermore, the α diversity of rare species tended to stabilize from moderate to extremely heavy grazing intensities, and there was a significant difference between MG and the control treatment ($P < 0.05$). The results revealed that although the rare species are inadequately represented in the community (*Jennifer Nagel et al., 2022*), they play a crucial role in the ecosystem and thus can be used as an important indicator for the assessment of the level of biodiversity (*Zavaleta, 2004*). The results of the correspondence analysis showed that the β diversity of the plant community and the α diversity of rare species responded strongly to different grazing intensities, which is consistent with the overall results of the present study.

## CONCLUSIONS

The results indicate that α diversity of all plant groups including dominant, common, and rare species, differed in response to grazing intensity, with that of rare species being the most sensitive index to grazing. As grazing intensity increases, most rare species decrease or even disappear, leading to a decrease in species diversity. Therefore, the α, β, and γ diversity of desert steppe species all initially decrease and then increase, with the β diversity showing the largest decrease. We conclude that rare species play an important role in assessing and maintaining biodiversity levels in a desert steppe. Studying the responses of rare species in desert steppes to different grazing intensities can effectively restore the plant species diversity of grazed areas and cope with the current pressures on the changing natural environment, to realize the sustainable use of grassland resources.

## ACKNOWLEDGEMENTS

We acknowledge the research team members for their invaluable assistance throughout the research process.

### Funding

This research was supported by the National Natural Science Foundation of China (31860156) and the Inner Mongolia Natural Science Foundation (2020MS03056). The funders had no role in study design, data collection and analysis, decision to publish, or preparation of the manuscript.

### Grant Disclosures

The following grant information was disclosed by the authors:
National Natural Science Foundation of China: 31860156.
Inner Mongolia Natural Science Foundation: 2020MS03056.

### Competing Interests

The authors declare there are no competing interests.

### Author Contributions

- Changlin Xue conceived and designed the experiments, performed the experiments, analyzed the data, prepared figures and/or tables, authored or reviewed drafts of the article, and approved the final draft.
- Shijie Lv conceived and designed the experiments, performed the experiments, authored or reviewed drafts of the article, and approved the final draft.
- Yanling Wu conceived and designed the experiments, performed the experiments, authored or reviewed drafts of the article, and approved the final draft.
- Jie Yun performed the experiments, prepared figures and/or tables, and approved the final draft.
- Rui Dong performed the experiments, prepared figures and/or tables, and approved the final draft.
- Wentao Wang performed the experiments, prepared figures and/or tables, and approved the final draft.

### Data Availability

The data and code are available in the Supplementary Files.

### Supplemental Information

Supplemental information for this article can be found online at http://dx.doi.org/10.7717/peerj.19087#supplemental-information.

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
