# Peer review of "Effects of different grazing intensities on plant species diversity at different spatial scales in a desert steppe in Inner Mongolia"

_PeerJ, doi:10.7717/peerj.19087_

## Round 0.1 · original submission · Major Revisions

After reviewing this revised version of your manuscript, I see that the main comments suggested by the reviewers have been included. However, there are still some details that need to be clarified before having a final version that can be published. Both reviewers have important comments that need to be addressed.

Reviewer 1 ·

Basic reporting

The article is well-structured and addresses an interesting topic for the conservation of natural grasslands under grazing in restrictive environments, such as desert steppes.

However, some adjustments need to be made for it to be suitable for publication. Specific points are detailed line by line. In general, it is necessary to improve the writing in English, reduce speculative aspects that do not directly arise from the obtained results, and clarify, for example, how grazing intensity was determined.

Experimental design

The experimental design is appropriate, as is the number of replicates and the statistical treatment of the data.

Validity of the findings

Although the work is not noted for its originality, as these aspects of the effect of grazing on the various components of diversity have been evaluated in other systems, it provides valuable information in a particular ecosystem.

Additional comments

The figures, tables, and supplementary material are appropriate.

Annotated reviews are not available for download in order to protect the identity of reviewers who chose to remain anonymous.

Reviewer 2 ·

Basic reporting

The MS deals with an important ecological topic, but the clarity and readability are clearly in need of improvement. The summary and introduction are convoluted, with numerous grammatical errors and awkward phrasing. The language should be revised by a native English speaker or a professional editor. The figures and tables are informative, but need better labeling and captioning to make them easier to understand. The inconsistent formatting of species names (e.g. missing author citations) needs to be corrected. The literature review is adequate, but could include more recent references to strengthen the context. Overall, the manuscript meets the basic structural requirements but needs to be refined for professional readability and consistency.

Experimental design

The experimental design is appropriate and in line with the research objectives as it focuses on the effects of grazing intensity on plant diversity at different spatial scales. However, some details need clarification: the method of randomization of sampling sites, the rationale for the size of the 1 m² plots, and the way in which grazing intensity was consistently controlled and monitored in each plot. The statistical analyzes are relevant, but the rationale for the choice of particular models (e.g. quadratic or exponential functions) should be explained in more detail. Ethical considerations and fieldwork permissions should be explicitly mentioned. Overall, the design is rigorous but requires more detail for reproducibility and transparency.

Validity of the findings

The results are supported by solid data collection and statistical analysis, but some interpretations require further elaboration. The non-linear response of diversity indices to grazing intensity is intriguing, but the ecological mechanisms behind these patterns should be better explained. The correspondence analysis effectively reveals relationships, but its implications for broader ecosystems could be discussed more clearly. The manuscript provides sufficient data to support its conclusions, but should strengthen the link between the results and their practical applications, particularly for conservation and sustainable grazing. Overall, the results are valid but could benefit from deeper contextualization and broader applicability.

Additional comments

Need certain improvements - see my comments on the pdf document itself.

Annotated reviews are not available for download in order to protect the identity of reviewers who chose to remain anonymous.

---

## Round 0.2 · accepted · Accept

After reviewing this revised version of your manuscript, I see that the comments suggested by the reviewers have been included. Therefore, I am satisfied with the current version and consider it ready for publication.

Reviewer 1 ·

Basic reporting

The authors have responded accurately and competently to my inquiries and have revised the manuscript in accordance to the suggestions provided. Therefore, I consider the article suitable for publication.

Experimental design

Correct.

Validity of the findings

Correct

Additional comments

Accept to be published